# Fuzzy Control Application to an Irrigation System of Hydroponic Crops under Greenhouse: Case Cultivation of Strawberries (Fragaria Vesca)

**DOI:** 10.3390/s23084088

**Published:** 2023-04-18

**Authors:** Edgar Maya Olalla, Andres Lopez Flores, Marcelo Zambrano, Mauricio Domínguez Limaico, Henry Diaz Iza, Carlos Vasquez Ayala

**Affiliations:** Facultad de Ingeniería en Ciencias Aplicadas, Universidad Técnica del Norte, Ibarra 100105, Ecuadoromzambrano@utn.edu.ec (M.Z.);

**Keywords:** intelligent control, hydroponic cultivation, fuzzy control, greenhouse

## Abstract

Hydroponics refers to a modern set of agricultural techniques that do not require the use of natural soil for plant germination and development. These types of crops use artificial irrigation systems that, together with fuzzy control methods, allow plants to be provided with the exact amount of nutrients for optimal growth. The diffuse control begins with the sensorization of the agricultural variables that intervene in the hydroponic ecosystem, such as the environmental temperature, electrical conductivity of the nutrient solution and the temperature, humidity, and pH of the substrate. Based on this knowledge, these variables can be controlled to be within the ranges required for optimal plant growth, reducing the risk of a negative impact on the crop. This research takes, as a case study, the application of fuzzy control methods to hydroponic strawberry crops (Fragaria vesca). It is shown that, under this scheme, a greater foliage of the plants and a larger size of the fruits are obtained in comparison with natural cultivation systems in which irrigation and fertilization are carried out by default, without considering the alterations in the aforementioned variables. It is concluded that the combination of modern agricultural techniques such as hydroponics and diffuse control allow us to improve the quality of the crops and the optimization of the required resources.

## 1. Introduction

Hydroponics generally refers to crops that do not use the soil to root the plants, and the essential nutrients for their growth can be supplied through the irrigation system [1,2]. The independence of the land use of hydroponic crops is an advantage for the benefit of spaces in which they are not suitable for crops due to soil characteristics or climatological variability. In this context, soil-less farming systems are considered an important technological component of modern greenhouses because of their advantages related to their wide variety of crops and ability to improve productivity and sustainability [3].

On the other hand, developing advanced control techniques whose applications are vast in various human activities has optimized the use of resources and improved the characteristics of the products obtained. Agriculture has not been exempt from applying these techniques [4,5]. Although fuzzy control techniques have been involved in a variety of cases for agriculture [6,7], there is still a need to carry out studies on applying these techniques to hydroponic crops under greenhouses using low-cost electronic devices to optimize irrigation systems.

The results obtained in the experimental part show that fuzzy control techniques generate control actions that time crop irrigation based on variables such as the ambient temperature, substrate temperature and humidity, pH, and electrical conductivity of the nutrient solution, resulting in a visible improvement in crop foliage and increased strawberry size compared to a timed irrigation system that does not consider the variables mentioned. The article is structured as follows: Section 2 describes the methods used to conduct the research; Section 3 details and analyzes the experimental implementation results; Section 4 gives the conclusions.

In recent years, research focused on precision agriculture has been proposed, especially in the automation of greenhouses with hydroponic applications. This section intends to find works related to the automation of hydroponic systems applying fuzzy logic:

The authors in [8] present a solution for improving the irrigation of rose crops. The objective is to reduce water consumption and operating costs, taking advantage of intelligent fuzzy logic controllers (FLCs) and environmental characteristics in the State of Mexico. They consider the main controllable variables that affect plant growth to be relative humidity (RH) and temperature (T). Through these variables, measured through sensors, each measurement is assigned a value on a scale of 0 to 255 decimal places for 8-bit digital circuits, where 0 is the value assigned to the minimum and 255 to the maximum. In addition, the temperature should be between 18 and 25 ${}^{\circ }$C, while the RH should be between 70 to 80%. Through the previous values, it is possible to control the position of a valve that supplies water, supplying an irrigation of 3 × 10${}^{3}$ mm of height per day. After implementing the FLC system, the water consumed between the months with the highest consumption was reduced by 15 L, while, for the rest of the year, the required flow rate is 7.5 L, thus achieving a reduction in the consumption of 0.2 L water per week.

On the other hand, [9] presents an experimental control that includes a manual adjustment that monitors the electrical conductivity (EC) and the required PH, and an automatic adjustment control that verifies if the designed function is fulfilled. Applying fuzzy logic, a regulation process of EC and PH is proposed for three different plants—lettuce, strawberry, and broccoli—in a hydroponic environment. In this experiment, 25 fuzzy rules were used.

In [10], the authors sought to estimate harvest labor, harvest time, post-harvest problems, storage, and transportation. In order to achieve this, they used WSN nodes and collected data through humidity and temperature sensors. This information was then taken to a Raspberry Pi, which has a connection to a database in the cloud. Through the machine learning processes used in this project (SVM, random forest, and naive Bayes), the algorithms made the best decision, and that information was downloaded into an app where the user is informed of the analyzed data. In this sense, it determines the precision of the machine learning algorithms used, where the SVM has a precision of 88%, the random forest has 70%, and, finally, the naive Bayes has 76%.

In some recent works, a nutritional control system for hydroponic lettuce crops was developed [11], where fuzzy logic was applied using environmental variables such as the total dissolved solids (TDSs) sensor, pH sensor, temperature sensor, and level sensor, among others. Optimal growth is achieved if the nutrient concentrations are between 500–800 ppm and the pH range is between 6.0 and 8.0. However, the authors do not show the growth stages of lettuce production.

In a similar work [12], an automatic monitoring and control system for environmental factors and the necessary nutrition for the cultivation of strawberries in vertical crops was developed. The system uses measurements of pH levels and an electrical conductivity (EC) sensor to determine the specific amount of nutrient solution based on fuzzy logic for three types of fertilizers: main fertilizer, flowering fertilizer, and fruiting fertilizer. Additionally, the system captures images of each strawberry plant using a camera navigation system and analyzes those images using a machine learning algorithm to identify the growth stage. The creation of a data set is specified, but it is not shared in order to visualize the growth stages.

In the work of [13], an increase of 40% in the production of the hydroponic cultivation of water spinach was proposed through the automation of nutrients using deep flow techniques (DFTs). A pH sensor was used to adjust the pH level of the nutrient solution. For future research, it is recommended to use the adaptive neuro-fuzzy inference system (ANFIS) to obtain a more adaptive fuzzy logic rule base and faster control system response results.

The authors of [14] proposed a feasibility study to implement an automatic pH regulation system in a domestic NFT hydroponic system for red lettuce crops. The variables involved in the monitoring of the fuzzy control system include pH, electrical conductivity, and volume sensors for the regulation of the pH and an on-off control of the water level, respectively. After obtaining the results, the authors suggest that further research should be carried out on the automation of hydroponic systems for different types of plants and the application of other pH-regulating substances that can be incorporated without causing harmful chemical reactions for the plants, such as nitric acid to acidify and potassium hydroxide to alkalize.

## 2. Methods

### 2.1. Greenhouse and Hydroponic Crops

A greenhouse is a specialized and artificially nuanced construction whose objective is to manage the parameters that intervene in the development of agricultural crops under controlled environments [15]. In general, the physical structure of greenhouses is built with light, transparent, and generally plastic materials.

The use of greenhouses, combined with the new techniques and cultivation systems, contribute to an improvement in agricultural production, in aspects related to an increase in production, quality improvement, pest control, etc. [16,17]. Greenhouses are classified according to factors such as the type of cover, materials of its structure, shape of the perimeter, and external environment, among others, the most common being structural and environmental parameters. In accordance with the above, the most common types of greenhouses are flat, chapel, sawtooth, tunnel, semi-Olympic, and asymmetric [15].

On the other hand, technological evolution has progressively led the greenhouse industry toward soil-less cultivation systems, commonly called hydroponic crops. Although the initial and operating costs are high compared to traditional crops, the advantages of hydroponic crops are framed in the reduction in production costs, harvest times, ease of supervision and control of environmental variables (humidity, temperature), independence of meteorological phenomena, reduction in spending on agricultural machinery and environmental pollution, etc. Hydroponics aims to grow plants without the use of agricultural land, supplying essential nutrients for plant development through artificial irrigation systems [18]. Modern hydroponic crops are characterized by a reduced water consumption, use of space, maximization of production, and improved quality of crops in terms of color, appearance, and absence of pesticides. These systems can be passive or active [19]. In passive systems, the roots of the plants are in direct contact with the nutrient substances (without recirculation). Active systems are more efficient and have a wide variety of configurations, such as reflux systems, the nutrient filtration technique (NFT), drip systems, aeroponics, floating roots, and wicks, among others.

An essential part of this type of cultivation technique is the substrates, which support plant growth, providing the ideal conditions for the root system to use nutrients properly. The substrates must comply with specific characteristics of porosity, capillarity, physical stability, and availability, as well as having a low cost, and they can be classified, according to the materials from which they come, into organic, inorganic, and artificial [2]. Organic substrates are waste products from the agricultural or industrial sector, such as sawdust, coconut fiber, rice husks, cereal straws, coffee husks, tree barks, etc. Inorganic substrates are composed of particles of more than 2 mm in diameter, such as pumice, gravel, volcanic rock, river sand, perlite, vermiculite, and expanded clay, among others. Finally, artificial substrates are obtained from synthetic polyethylene foam and foamy agricultural materials [20].

### 2.2. Sensorization

Currently, sensors are an indispensable tool for monitoring environmental conditions in greenhouses. The main variables to identify are temperature and humidity, of both the soil and the substrates. Additionally, in hydroponic crops, it is necessary to measure variables such as the pH and electrical conductivity of the circulating nutrient solution.

The environmental temperature directly influences the crops. An adequate control of the temperature (e.g., through an air conditioning system correctly adjusted to the particular requirements of the crop in question) manages to improve production safely, guaranteeing ideal climatic conditions throughout the year [21]. The most widely used temperature sensors in greenhouses are resistance (RTD) and thermistors that do not exceed 0.5 ${}^{\circ }$C of imprecision [22].

To monitor the humidity of the substrate, it is necessary for the humidity sensor developed to be based on a corrosion resistance. There is a wide variety of soil humidity sensors, but the most widely used are those that work under the principle of electrical resistance and capacitance [23].

The acidity or alkalinity of water is measured by the potential of hydrogen (pH). Crops grown in optimal conditions, specifically hydroponics, require a precise control of the pH of the nutrient solution, since the use of nutrients by the plants depends on this factor; otherwise, the substrate becomes an insoluble saline solution that the crop cannot use. The measurement range for pH varies between 0 and 14. When this value is greater than 7, a basic solution is obtained; otherwise, it is acidic. Several methods can be used for pH measurement, including the glass electrode system and the ISFET transistor [24,25].

Sensors measure the quality of the solution to conduct an electrical current between two electrodes, usually encapsulated in a probe [24]. Electrical conductivity makes it possible to determine the concentration of soluble salts in the nutrient solution of hydroponic crops. A high electrical conductivity indicates a higher salt concentration [25].

### 2.3. Fuzzy Control

Fuzzy control systems simulate human reasoning (artificial intelligence) for the efficient management of complex processes. In practice, one of the advantages of fuzzy controllers lies in their ease of designing and modeling, in natural language, using fuzzy rules, complex specifications of the processes to be controlled [26]. Fuzzy logic makes modeling possible in ambiguous terms. The variables involved in the processes, through a set of logical “If-Then” rules, are based on the knowledge and experience of experts in the specific processes to be automated.

The fundamental concepts that are part of fuzzy logic and its relation to controller design are briefly summarized below [27,28]. Fuzzy sets are tools that allow for modeling processes and/or concepts whose limits are not well defined; for example: age groups, lengths, weights, etc. Formally, a fuzzy group (subset) is characterized within an operative domain X by a membership function A defined in a range of [0, 1], which associates to each element of the domain a value that represents the degree of membership from that element to that set:(1)μA(x):X→[0,1]
(2)A={(x,μA(x)),x∈X}
(3)μA(x)=0,∀×doesnotbelongtotheset
(4)μA(x)=1,∀×belongsentirelytotheset

Fuzzy logic can be used to design controllers applicable to various systems. The operation of these controller is based on prior knowledge and information provided by one or more reliable sources. The easiest way to use a fuzzy controller is to establish a system under a closed loop (feedback control), as shown in Figure 1. The fuzzy controller determines the actions to perform based on the error obtained by comparing the value of the variable, which is taken as a reference with the value measured by the sensor. In general, a fuzzy controller consists of a quantization (blurring) stage, a rule base, an inference engine, and a translation (debunking) stage, as shown in Figure 2. For this specific case, the action control is applied to the irrigation system of the hydroponic cultivation of strawberries.

The blur stage converts a numeric value into a fuzzy set that represents it; the most common methods are single point, Gaussian blurring, and triangular blurring [27,28]. The rule base contains all of the information about the application to be controlled and the objectives of the controller; it is composed of a database and the set of linguistic rules used to govern the variable. The database provides the definitions for the manipulation of the fuzzy data and the rules characterize the control policies and objectives that the experts use to carry out the control. The inference engine determines the corresponding control action for each stage of the process to be controlled. The strategy for managing the database includes support for fuzzy set definitions. Blurring reconverts the blurred set to the expected numerical value; it consists of a mathematical process where the conversion is performed to a non-fuzzy value, which can be understood. The methods used to obtain this value are the center of gravity method and averaged centers, among others.

## 3. Experimentation, Analysis of Results, and Advantages of the Proposal

### 3.1. Greenhouse and Irrigation System

For the experimentation, a greenhouse was built in the parish of Quiroga, located in the Cotacachi canton of Imbabura province in Ecuador, and the San Andreas strawberry variety was cultivated. The greenhouse is located at an altitude of 2418 m above sea level, with an annual temperature variation of 8 ${}^{\circ }$C to 24 ${}^{\circ }$C. Given these environmental conditions, the chapel-like greenhouse is the one selected to be built for its application characteristics in temperate and cold climates. Figure 3 shows a diagram of the front view of the greenhouse, whose dimensions are 4 m long, 4 m wide, and 3 m to the highest point; the whole front part is intended for ventilation, and four containers for the irrigation system of the strawberry crop are implemented. Figure 4 shows the actual construction of the greenhouse.

The irrigation system is divided into two parts. The first part consists of the first two containers used to implement a traditional timed irrigation system. The second part comprises the last two containers in which an irrigation system uses fuzzy control techniques. A schematic diagram of the irrigation system for the first two containers is shown in Figure 5; the system for the remaining containers is of similar characteristics and design. The system consists of a 0.5 hp electric pump responsible for circulating the nutrient solution stored in the tank. Additionally, the system consists of ball wrenches whose function is to minimize pressure generated by the pump and regulate the flow of water to the containers. The solenoid valves of the system allow for the control of the water passage to the pipes; depending on the control actions, the flow of the nutrient solution or simple water from the residential drinking water will allow for circulation. Finally, a system that returns the nutrient solution to the tank is implemented using gravity in order to have a degree of inclination.

### 3.2. Data Acquisition and Transmission System

In the Figure 6 and Figure 7 shows the block diagram of the general operation of the system, where the connection of each of the parts that make up the data acquisition system and the controllers implemented for smart irrigation can be seen. Due to the electronic features and simple low-cost tools, two Arduino boards were used for data acquisition, information transmission, and the implementation of the controllers. The Arduino Uno board with an AT-mega328P microcontroller acquires signals from ambient temperature, conductivity, pH, substrate temperature, and humidity sensors. In addition, a fuzzy control for the irrigation of water without nutrients was implemented in this microcontroller. For communication between microcontrollers, Zigbee technology was used using additional modules that implement this technology. The second board is Arduino Mega, in which, a fuzzy controller was implemented for irrigation with water with nutrients.

#### Sensors

The ambient temperature sensor used was the DHT22, which has a measuring range of 40 ${}^{\circ }$C to 80 ${}^{\circ }$C, with a rank of precision of ±0.5 ${}^{\circ }$C.For the humidity measurement of the substrate, a capacitive-type humidity sensor SKU SEN0193 was used, which has corrosion resistance characteristics and is compatible with Arduino.The ds18b20 sensor was used to measure the temperature of the substrate, as it highlights characteristics such as its small size (6 mm in diameter, 3 cm long) and low energy consumption.The pH sensor used was the Atlas Scientific Lab Grade pH Probe, whose accuracy is ±0.1 pH, and it can be continuously immersed in the nutrient solution.The sensor used to measure electrical conductivity was the a1003v1 model, which can be submerged the entire time by continuously making measurements.

Figure 8 shows the mounting of the pH (a) and electrical conductivity (b) sensors on the nutrient-rich substance circulating in the irrigation system. Figure 9 shows the arrangement and assembly of the ambient temperature, humidity, and substrate temperature sensors.

For a better management of the experimental data and control actions generated by the controllers, a zoning consisting of two zones was implemented. Zone 1 includes the cultivation of strawberries and zone 2 corresponds to the place where the management, interpretation, and visualization of the information obtained from each sensor.

### 3.3. Control System

The overall control system consists of two fuzzy controls. Fuzzy controller 1 generates control actions that regulate the water irrigation time with nutrients to the crop based on information from the substrate humidity, substrate temperature, and ambient temperature sensors. Fuzzy controller 2 is responsible for determining the irrigation time for water without nutrients. The input variables for this controller are electrical conductivity and pH. Figure 10 and Figure 11 show the distribution of variables for controllers 1 and 2, respectively.

Unlike traditional soil crops, semi-hydroponic crops supply essential elements such as nitrogen, phosphorus, potassium, sulphur, calcium, and magnesium, which are considered as macronutrients, as well as micronutrients such as zinc, manganese, copper, iron, molybdenum, boron, chlorine, and nickel, which are incorporated through the irrigation system.

For the experiment, Hakaphos was used as a nutrient solution in three different types, depending on the requirements of each stage of growth.

Hakaphos Violet NPK N-P2O5-K2O 13-40-13 with boron, copper, iron, manganese, molybdenum, and zinc is indicated for crop rooting.

Hakaphos Red 18-18-18 NPK with free micronutrients, a low concentration of chloride (Cl 1%), and a lack of sodium and urea is indicated for flowering and balanced crop growth.

Hakaphos Yellow 5-17-19 NPK with magnesium, sulfur, micronutrients, and a high NK content is a reduced-phosphate nutrient salt for fertilization with an almost balanced NK ratio and a slight emphasis on ammonium. It is indicated for growth and the beginning of fruit production.

#### 3.3.1. Fuzzy Sets

The I–V boxes then define the fuzzy sets and ranges for each monitored variable. The fuzzy output sets of both controllers are shown in box VI.

#### 3.3.2. Fuzzy Rules

These rules are generated from all possible combinations that can be made between the fuzzy subsets. The Matlab software was used to create these combinations, where the full assigned name of each set and subset fuzzy is used. In order to simplify this and facilitate the understanding of each rule, it was decided to assign initials to each set for both controllers.

For fuzzy controller 1, there were 75 rules formed from the combination of the fuzzy subassemblies of each set value of environment temperature ( Table 1), substrate temperature (Table 2) and substrate humidity (Table 3 and Table 4). For fuzzy controller 2, 20 rules were established considering the combination of the blurred subsets of electrical conductivity and pH.

#### 3.3.3. Controller Test

Fuzzy controller 1 generates control actions that depend on the substrate humidity, substrate temperature, and ambient temperature. The temperature and humidity of the substrate should be kept below 25 ${}^{\circ }$C and 550, respectively. Note that, for ease, the reported values of humidity correspond to the values of the Arduino digital–analog converter, where the minimum value corresponds to a relative humidity of 0% and the maximum value to 100%. Given an ambient temperature inside the greenhouse of 26 ${}^{\circ }$C, substrate temperature of 27 ${}^{\circ }$C, and humidity value of 663, the fuzzy controller designed in Matlab generates an irrigation time (activation of the electric pump) of 502 s, as shown in Figure 12.

Figure 13 shows that, after the electric pump is activated, in approximately 2 min, the temperature begins to drop rapidly from 27 ${}^{\circ }$C to 22 ${}^{\circ }$C, and thereafter keeps fluctuating between 20 ${}^{\circ }$C and 23 ${}^{\circ }$C. The value shown by the controller is 502 s, so if we consider the time when the electric pump starts to work (9:17:14), the result is 9:25:17. This graph shows additional time to evidence that the temperature values of the substrate remain below the limit, which is 25 ${}^{\circ }$C. Table 5 shows the fuzzy sets established for the pH of the culture, while Table 6 shows the corresponding outputs of the fuzzy controllers. In the same way, Figure 14 shows that, once the electric pump is activated, in approximately 2 min, the scale of the analog values begins to descend rapidly from 663 to 500. This scale can be extended up to 1024, which is the maximum value that can be reached. Regarding the analog values, if they are closer to this value, it means that the humidity decreases and irrigation is needed; the next time, the scale is changed until it reaches 400. Then, this value continues to fluctuate between 380 and 420.

The variables of electrical conductivity and pH are the input of fuzzy controller 2. Given an electrical conductivity value of 1113 μS/cm and a pH value of 5, 2, the control action generated is to activate the electric pump for 310 s, as seen in Figure 15, performed in Matlab. On the other hand, Figure 16 shows that, after activating the electric pump, in approximately 5 min, the electrical conductivity begins to drop until reaching an approximate value of 900 micro-siemens. Unlike the variables of the temperature and humidity of the substrate, this parameter is measured at the end of the PVC pipes. In addition, even after waiting an extra 300 s until the water with the nutrient solution from controller 1 returns to the tank, some residues remain in the substrate; therefore, until the conductivity value begins to drop, the time is greater. Thereafter, the value continues to drop until after 10:12 min, where it remains at an approximate value of 800 micro-siemens. Figure 17 shows that the PH values keep fluctuating most of the time in a range of values from 5 to 6, where the peaks shown are caused by a voltage drop in the sensor. 

## 4. Conclusions

Hydroponics is an agricultural method that, in theory, can be applied to any type of plant. This method is based on the use of circulating water, which contains minerals that are absorbed by the roots of plants. Hydroponics is gaining more and more acceptance because of the many advantages that it offers, among which, is a greater productivity. When implemented in greenhouses, it significantly contributes to the improvement of agricultural production in aspects such as an increased quantity, improved product quality, and pest control, among others.

The objective of this research allows for automating a hydroponic strawberry cultivation (Fragaria vesca) in a greenhouse through a diffuse control system. The fuzzy logic is achieved through rules and fuzzy sets that reinforce the actions of the controller based on the input variables of pH, electrical conductivity, temperature, and humidity of the substrate, allowing the electronic system to have the capacity to supply the necessary amount of water to strawberry plants at time intervals for cultivation.

Unlike traditional soil crops, hydroponic crops supply essential elements such as nitrogen, phosphorus, potassium, sulfur, calcium, and magnesium, which are considered as macronutrients, and micronutrients such as zinc, manganese, copper, iron, molybdenum, boron, chlorine, and nickel. For experimentation, three types of the Hakaphos nutrient solution were produced according to the requirements of each stage of growth and incorporated through the irrigation system.

This automated irrigation system improved the crop quality, evidenced by an increased plant growth, foliage, and fruit size compared to an automated irrigation system with a fixed timer setting. This approach may have broader applications in agriculture and horticulture in general, indicating the need for the continued research and development of new techniques and tools to improve the crop productivity and efficiency.

Finally, in Appendix A and Appendix B, the differences between the two systems are shown: the Timed System (Figure A1) and the Fuzzy Controller System (Figure A2), during the growth process until the fruit reaches maturity.

## Figures and Tables

**Figure 1 sensors-23-04088-f001:**
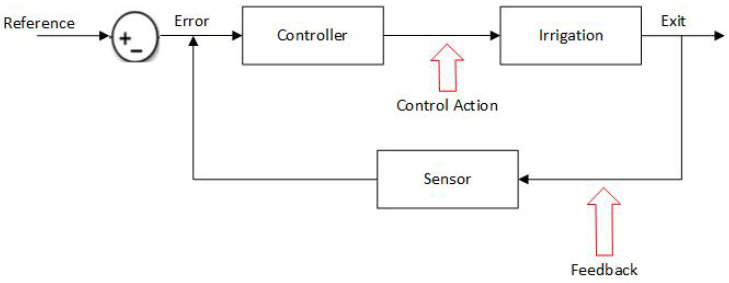
Feedback control system.

**Figure 2 sensors-23-04088-f002:**
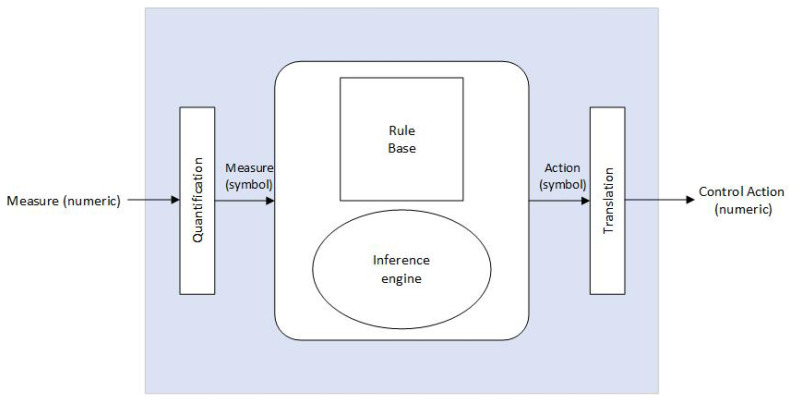
Structure of a fuzzy controller.

**Figure 3 sensors-23-04088-f003:**
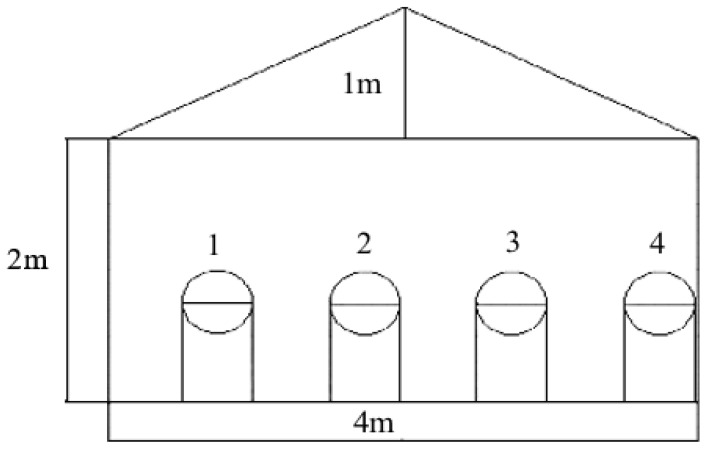
Front view of the greenhouse.

**Figure 4 sensors-23-04088-f004:**
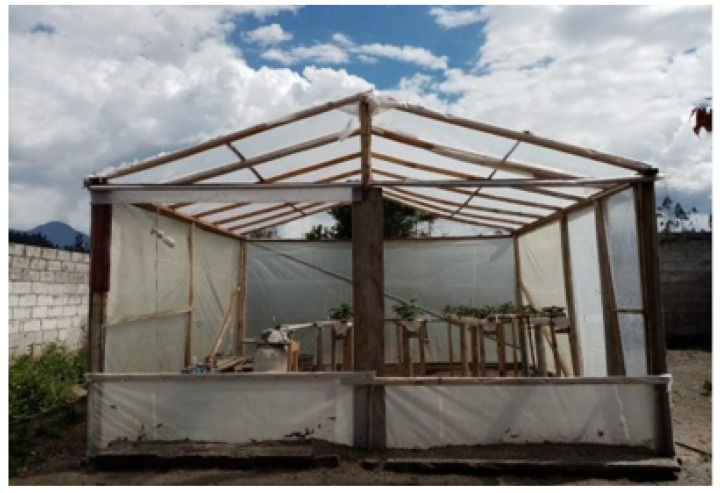
Chapel-type greenhouse.

**Figure 5 sensors-23-04088-f005:**
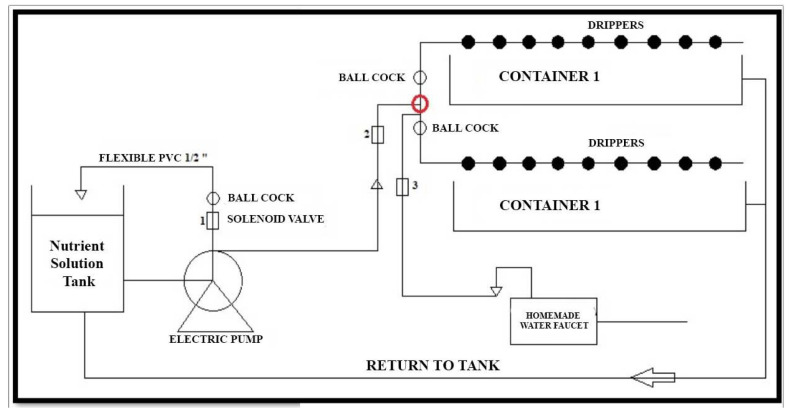
Implemented irrigation system scheme.

**Figure 6 sensors-23-04088-f006:**
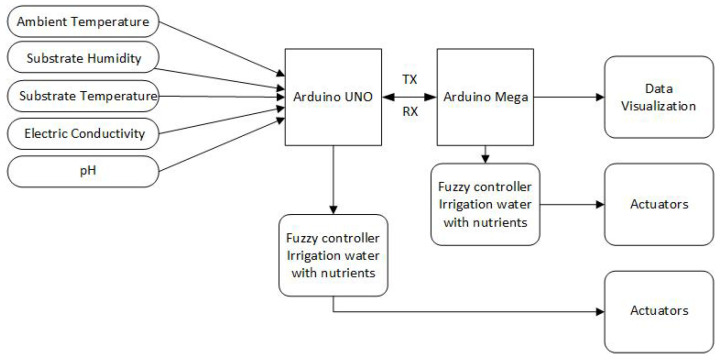
System architecture.

**Figure 7 sensors-23-04088-f007:**
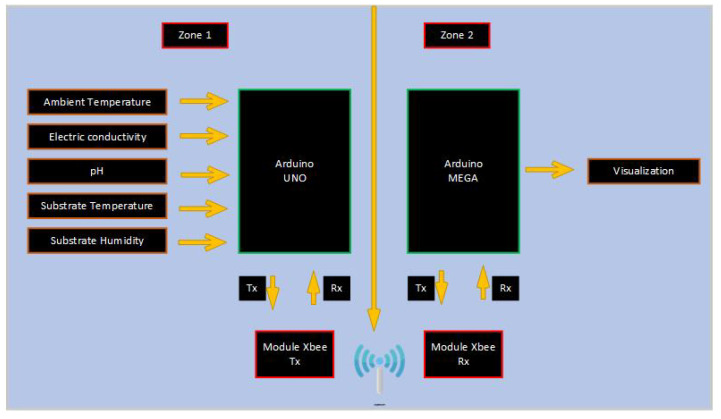
System zonification.

**Figure 8 sensors-23-04088-f008:**
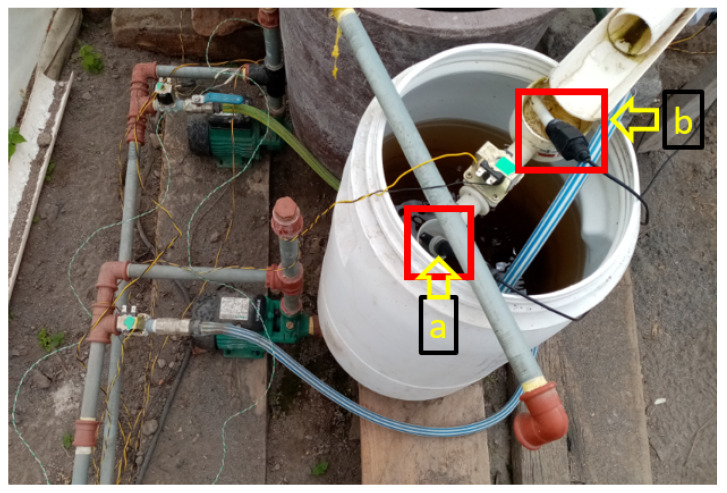
(**a**) Probe for pH; (**b**) probe for electrical conductivity.

**Figure 9 sensors-23-04088-f009:**
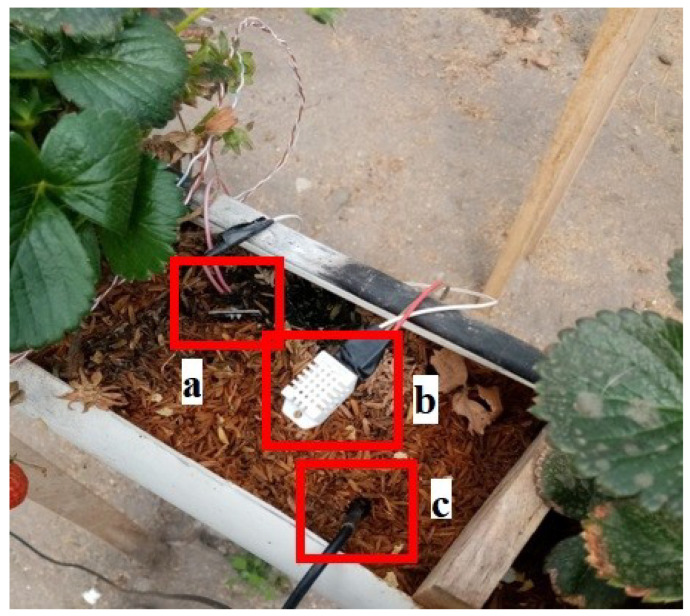
(**a**) Substrate humidity sensor, (**b**) dht22 sensor, (**c**) ds18b20 probe.

**Figure 10 sensors-23-04088-f010:**
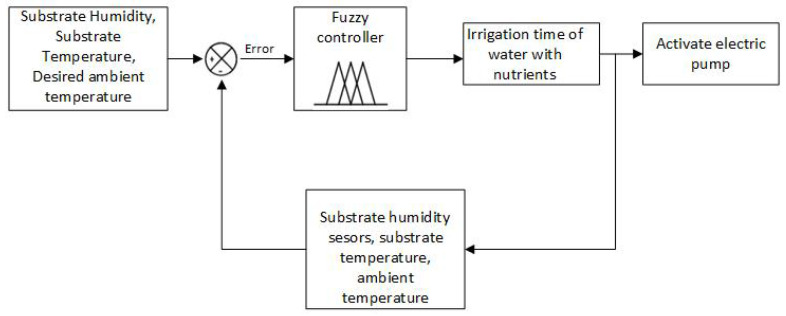
Fuzzy controller 1.

**Figure 11 sensors-23-04088-f011:**
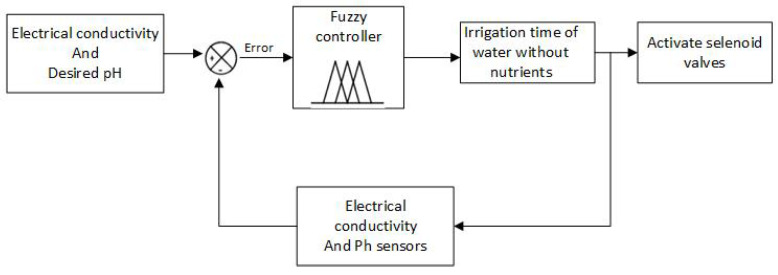
Fuzzy controller 2.

**Figure 12 sensors-23-04088-f012:**
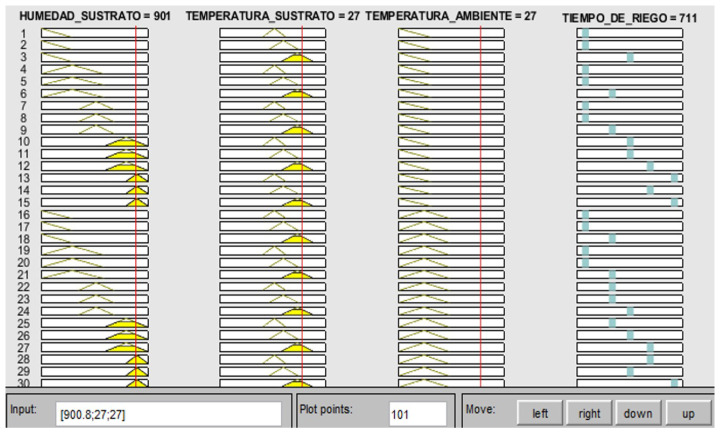
Fuzzy controller 1 results.

**Figure 13 sensors-23-04088-f013:**
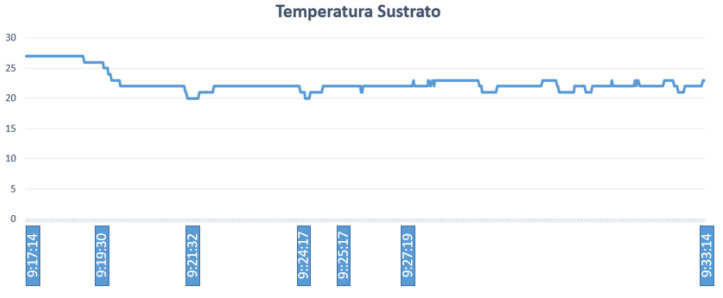
Substrate temperatura variation.

**Figure 14 sensors-23-04088-f014:**
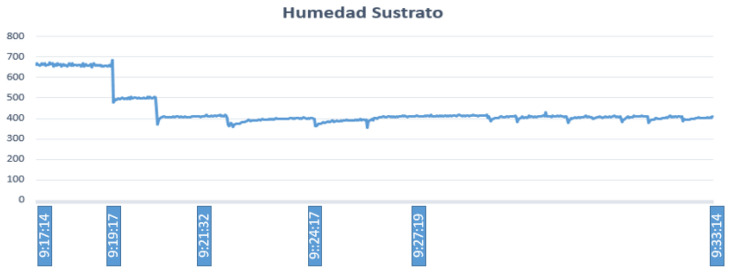
Substrate humidity variation.

**Figure 15 sensors-23-04088-f015:**
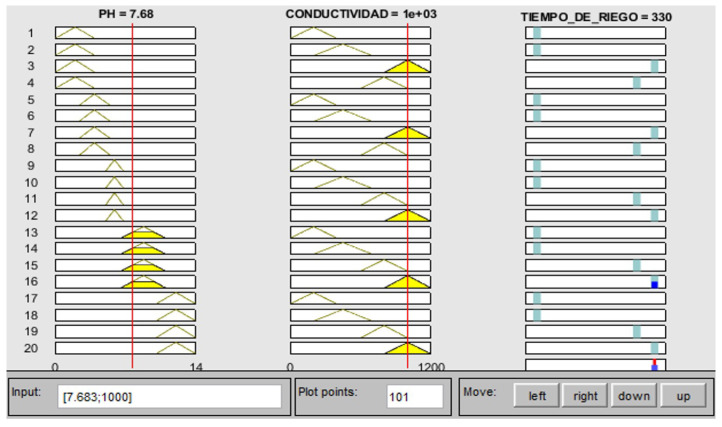
Fuzzy controller 2 results.

**Figure 16 sensors-23-04088-f016:**
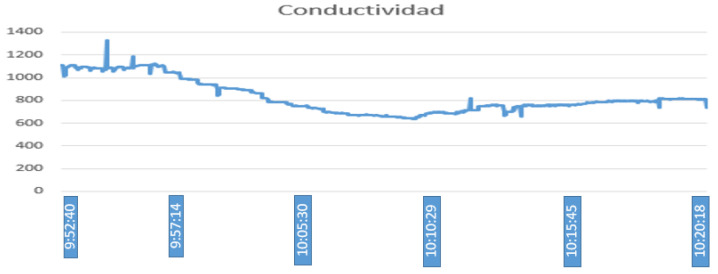
Nutrient substance Electrical conductivity.

**Figure 17 sensors-23-04088-f017:**
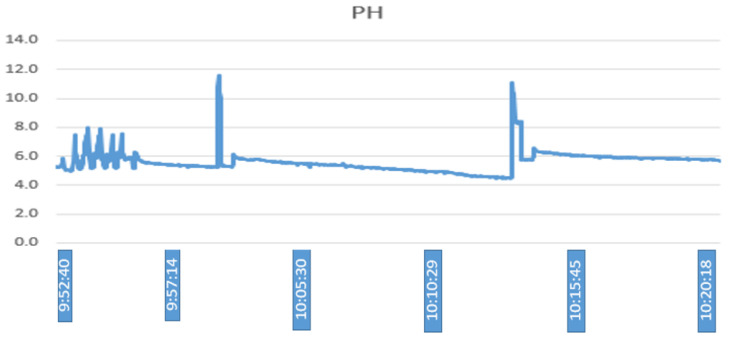
Nutrient pH.

**Table 1 sensors-23-04088-t001:** Ambient temperature fuzzy sets (${}^{\circ }$C).

	Very Cold	Cold	Temperate	Hot	Very Hot
Range [0–32]	[0,11]	[0,17]	[12,20]	[17,27]	[23,32]

**Table 2 sensors-23-04088-t002:** Substrate temperatura fuzzy sets (${}^{\circ }$C).

	Semi-Optimal	Optimal	Hot
Range [0–31]	[14,22]	[16,26]	[20,31]

**Table 3 sensors-23-04088-t003:** Substrate humidity fuzzy sets.

	Too Moist	Very Moist	Moist	Dry	Very Dry
Rank (0–1024)	[0,300]	[300,600]	[350,700]	[600,1024]	[800,1024]

**Table 4 sensors-23-04088-t004:** Electric conductivity fuzzy sets.

	Little Saline	Moderate Saline	Saline	Too Much Saline
Rank (0–1000)	[0,400]	[200,700]	[600,1000]	[800,1200]

**Table 5 sensors-23-04088-t005:** PH fuzzy sets.

	Very Acidic	Acidic	Normal	Alkaline	Very Alkaline
Range (0–14)	[0,4]	[2.4,5.5]	[5,6.8]	[6.5,11]	[10,14]

**Table 6 sensors-23-04088-t006:** Controller outputs fuzzy sets.

	Irrigate Nothing	Irrigate Little	Irrigate Middle	Irrigate Much	Irrigate Too Much
Controller 1 Range (0–796)	[0]	[0,240]	[0,400]	[0,580]	[0,796]
Controller 2 Range (0–330)	[0]	[0,120]	[0,180]	[0,280]	[0,330]

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
