# Peer review of "Fuzzy Control Application to an Irrigation System of Hydroponic Crops under Greenhouse: Case Cultivation of Strawberries (Fragaria Vesca)"

_sensors, 2023, doi:10.3390/s23084088_

Round 1

Reviewer 1 Report

Various sorbents and various compositions of nutrient media were used in the work. The authors are required to give a description and chemical composition of the sorbents and nutrient media used in the section "Objects and Methods of Research"

Author Response

According to the suggestions made, changes to the paper were implemented.
A general review of the entire document was carried out with regard to grammar and language expressions.
The quality of the figures was improved.
The state of the art and the methodology were reviewed. The description of the general concepts of fuzzy control and the methods used during the development of the investigation was improved.

Reviewer 2 Report

The paper describes a study of the use of fuzzy control techniques in hydroponic agriculture for the growth of strawberry crops. The technique involves using specific irrigation systems that provide the necessary nutrients to the crops based on information gathered from various variables such as temperature, humidity, pH, and electrical conductivity. The application of fuzzy control techniques resulted in larger foliage and fruit sizes compared to fixed irrigation systems. Overall, the paper was well-written and quite clear in its methodology. The authors presented adequate results and discussion for scientists in this matter, using it as a basis for future research and development. However, there are some comments to improve the quality of this manuscript listed below:

1. Abstract: Should mention the background of the study, objectives of the study, method used, major findings, and the concluding remarks.

2. Introduction: Please include additional related research.

3. The article should add a literature review or additional references.

4. A more detailed analysis of the results obtained has to be added.

5. Please check for typographical and grammatical errors.

6. Here is a reference relevant to your area. You may use.

- Phasinam, K., Kassanuk, T., Shinde, P.P., Thakar, C.M., Sharma, D.K., Mohiddin, M.K., & Rahmani, A.W. (2022). Application of IoT and Cloud Computing in Automation of Agriculture Irrigation. Journal of Food Quality, 2022, 8285969. DOI: 10.1155/2022/8285969.

I wanted to take a moment to provide you with some feedback on your recent research paper on hydroponic crops and fuzzy control techniques.
         1. The main question addressed in the research is the impact of fuzzy control techniques on hydroponic agriculture for the growth of strawberry crops.

         2. The topic is relevant and addresses a gap in the field of hydroponic agriculture.

         3. The study adds to the subject area by showing the positive impact of fuzzy control techniques compared to fixed irrigation systems.
         4. To improve the methodology, the authors could consider a larger sample size, controlling for other environmental factors, and comparing the results with those of traditional hydroponic techniques.

         5. The conclusions are consistent with the evidence and arguments presented.

         6. The references appear to be appropriate and relevant to the subject area.

         7. The figures should be checked for image clarity to ensure the results are clearly presented and interpretable.

         Thank you for your contributions to the field of hydroponic agriculture. I hope this feedback is helpful for the authors in their future research endeavors.

Author Response

(The authors gave the same response as above.)

Reviewer 3 Report

The introduction is too short and lacks proper problem formulation. The introduction should contain some previous works' short descriptions so that readers can understand what the problem is and how other researchers tackled the problem. This way the authors formulate the problem and discuss some existing solutions and propose their solution. Therefore, the introduction section needs to be re-written.

Figures 4, 6 and 7 may not be required. These figure can be omitted.

The texts before and after a figure are not understandable. Perhaps, some texts were misplaced due to figure insertion. This must be fixed.

The authors showed many figures in the result sections and they only mentioned that this figure shows this and as such. However, an analysis of the result is missing. There must be an analysis of each result figure. 

There are some comments in the attached file also.

Author Response

(The authors gave the same response as above.)

Author Response

(The authors gave the same response as above.)

Round 2

Reviewer 3 Report

The abstract still has 2 paragraphs. I do not think it is ok to have 2 paragraph abstract in a journal paper.

The conclusion is too short. The conclusion should repeat some introduction and methodologies such as what the problem was and how it was tackled. Finally, the results are to be mentioned. The authors kept only the results in the conclusion. So, I think, the conclusion needs to be improved.

In line 297, %100 was written. I think it would be 100%.

In the results, the figures are just given and very little analysis is done. A discussion on what Fuzzy controllers are actually doing is needed. In the appendix, the figures show that fuzzy controllers worked well but the figure 13,14,16,17 is not much understandable. These figures need descriptions. Also, can you show differences through these figures as you have shown in the figures of the appendix?

Author Response

Based on suggestions made by reviewers in this second round, the following changes have been made to the document:
- Consolidated the entire summary into a single paragraph.
- The conclusions were improved, including a description of the investigation and its characteristics.
- Fixed misspelling of line 297.
- The descriptions of figures 13, 14, 16 and 17 were introduced.

Reviewer 4 Report

Manuscript can be published in its current form

Author Response

(The authors gave the same response as above.)

Round 3

Reviewer 3 Report

The manuscript can be published without any more modification